# How is enrollees' trust in health insurers associated with choosing health insurance?

**Frank J. P. van der Hulst**[1]*, **Anne E. M. Brabers**[1], **Judith D. de Jong**[1,2]

**1** Nivel, The Netherlands Institute for Health Services Research, Utrecht, The Netherlands, **2** Maastricht University, Maastricht, The Netherlands

* f.vanderhulst@nivel.nl

**Data Availability Statement:** The minimal anonymized data set necessary to replicate our study findings are uploaded to EASY (https://doi.org/10.17026/dans-x5m-hppm). The data set may only be used under the conditions laid down in the

## Abstract

In a healthcare system based on managed competition, health insurers are intended to be the prudent buyers of care on behalf of their enrollees. Equally, citizens are expected to be critical consumers when choosing a health insurance policy. The choice of a health insurance policy may be related to trust in the health insurer, as enrollees must believe that the health insurer will make the right choices for them when it comes to purchasing care. This study aims to investigate how enrollees' trust in health insurers is associated with their choice of a health insurance policy in the Netherlands. We will focus on the switching behaviour of enrollees and the choice of a policy with restrictive conditions. In February 2022, a questionnaire was sent to a representative sample regarding gender and age of the adult Dutch population. In total 1,125 enrollees responded, a response rate of 56%. Respondents were asked about the choices they made in choosing health insurance. Trust in health insurers was measured using the Health Insurer Trust Scale (HITS), a validated multiple item scale. Descriptive statistics, a paired t-test and logistic regression models were conducted to analyse the results. Of all respondents, 35% indicated that they agree, or completely agree, with the statement that they trust health insurers completely. In addition, trust in enrollees' own insurer is slightly higher than trust in other insurers (36.29 vs. 33.59, p<0.001). Furthermore, we found no significant associations between trust in health insurers, and whether enrollees have either switched health insurers or have chosen a policy with restrictive conditions. This study showed that enrollees' trust in health insurance in the Netherlands is relatively low and that trust in their own insurer is slightly higher than trust in other insurers. Furthermore, this study does not show a relationship between trust in health insurers and, either switching health insurers, or choosing a policy with restrictive conditions. Nevertheless, attention for increasing the trust in health insurers might still be important, as low trust may have negative consequences for other elements of the functioning of the healthcare system.

## Introduction

In recent decades several countries such as the Netherlands, Germany, and Switzerland, have switched to a healthcare system based on managed competition [1–3]. This is a system

privacy regulations of the Dutch Health Care Consumer Panel. Therefore, the dataset is available on request. Data requests will be assessed against these regulations. Additionally, the same minimal anonymized data set is available upon request from prof. Judith D. de Jong (j.dejong@nivel.nl), project leader of the Dutch Health Care Consumer Panel, or the secretary if this panel (conusmentenpanel@nivel.nl). The Dutch Health Care Panel had a program committee, which supervises processing the data of the Dutch Health Care Consumer Panel and decides about the use of the data. This program committee consists of representatives of the Dutch Ministry of Health, Welfare and Sport, the Health Care Inspectorate, Zorgverzekeraars Nederland (Association of Health Care Insurers in the Netherlands), the National Health Care Institute, the Federation of Patients and Consumer Organisations in the Netherlands, the Dutch Healthcare Authority and the Dutch Consumers Association. All research conducted within the Consumer Panel has to be approved by this program committee. The committee assesses whether a specific research fits within the aim of the Consumer Panel, which is to strengthen the position of the health care user.

**Funding:** The data collection of this study was funded by the Dutch Ministry of Health, Welfare and Sport. The funders had no role in study design, data collection and analysis, decision to publish, or preparation of the manuscript.

**Competing interests:** The authors have declared that no competing interests exist.

regulated by the government that stimulates competition between third party purchasers, which in most cases are health insurers, and healthcare providers [1, 2, 4, 5]. Health insurers play an important role in such a healthcare system. They are intended to behave as the prudent buyers of care on behalf of their enrollees [1, 4, 5]. The possibility that enrollees may switch to a competitor is supposed to be a key element in creating competition between health insurers. Since enrollees are able to switch health insurers every year, this should give health insurers the incentive to keep enrollees satisfied by offering the most attractive health insurance package. This may include favourable conditions and prices to attract new enrollees and retain affiliated ones [1, 4–6]. During the purchasing negotiations with healthcare providers, according to the theory, health insurers, therefore, aim to improve the quality of care and reduce prices [5–7]. By doing this they should improve their competitive position in relation to other health insurers. In addition to competition between health insurers, the system of managed competition should stimulate competition between healthcare providers. In some countries, health insurers are able to apply selective contracting of care providers [4, 6, 8]. This means that health insurers only conclude contracts with providers with whom they have been able to reach reasonable agreements regarding costs, volume, and quality of care. As a result, healthcare providers might be concerned that they will lose market share if they are not contracted by health insurers. In order to make selective contracting effective, health insurers must be able to steer enrollees towards their contracted providers. Restrictive health insurance policies play an important role in this. If enrollees have such a health insurance policy, then they have to make a co-payment if the healthcare provider treating them does not have a contract with their insurer[6]. This makes it financially less attractive for enrollees to visit these providers.

## Trust in the health insurer

In countries with a healthcare system based on managed competition, citizens are expected to be critical consumers when choosing a health insurance policy. The choice of a health insurance policy may be related to trust in the health insurer. Trust can be defined as "the optimistic acceptance of a vulnerable situation in which the truster believes the trustee will care for the truster's interests" [9]. Theory regarding consumer behaviour states that trust is required when there is uncertainty. And too that the higher the initial perception of risk, the higher the trust required to facilitate transactions [8, 10]. Trust is closely related to the truster's risk arising from the uncertainty about the trustee's motives, intentions, and future behaviour [8, 11]. Furthermore, trust has been shown to reduce perceived risk [8, 10, 12].

This uncertainty exists with regard to the choice of a health insurance policy. For example, enrollees do not have experience with all care providers, so they cannot be certain about the quality of care provided by different providers. In addition, enrollees have no certainty about what care they will need in the future. The risk derived from this uncertainty may lead to the situation that enrollees need to trust their health insurer is acting in their best interests. Enrollees who trust their health insurer believe that they will make the right choices for them when it comes to purchasing care [13].

In this study, we investigate how enrollees' trust in health insurers is associated with their choice of a health insurance policy in the Netherlands. There are ten health insurance groups, nineteen health insurers, sixteen labels, and one authorised representative in the Netherlands in 2022 [14]. Every year, from November 12, enrollees have the option of switching to another health insurer or health insurance policy for the coming year. The last day to decide to discontinue a health insurance policy is December 31. The last day to enroll in a new health insurance is January 31. The new health insurance then starts January 1, retroactively if enrollees enroll in a new health insurance policy in January. Different health insurers and labels offer around

60 different types of health insurance policies for the basic health insurance in 2022 [14]. Although all insurers offer the same extensive basic insurance, of which the content is determined by the government, these policies usually differ in their terms of conditions. For instance, policies may differ in the degree of freedom of choice of healthcare providers as care from non-contracted providers is not always fully reimbursed.

Dutch enrollees have the least choice of fully reimbursed healthcare providers if they opt for health insurance policies with restrictive conditions [15]. These, according to the definition of the Dutch Healthcare Authority (NZa), are a group of policies for which the reimbursement rate is below 75% for the use of non-contracted care. These health insurance policies are, in general, available at a lower premium compared to those without restrictive conditions [15]. If enrollees have such a health insurance policy, their choice of healthcare providers is limited due to selective contracting if they do not want to face extra, out-of-pocket, payments. Trust in the health insurer is expected to play a role in the willingness of enrollees to allow their health insurer to limit their choice of providers. This has also been shown in previous research. A study by Bes et al. shows that when enrollees' trust in the health insurer is lower, they are less likely to accept selective contracting [8]. However, the association between trust in the health insurer and enrollees' choices with regard to a health insurance policy has not been studied before.

This article, therefore, focuses on how enrollees' trust in health insurers is associated with the choices they make with regard to health insurance policies. We examine three research questions. What is the level of enrollees' trust in health insurers, and does it differ between their own and other health insurers? Secondly, what is the relationship between trust in health insurers and enrollees' switching behaviour? And, thirdly, what is the relationship between trust in health insurers and enrollees' choice of a restrictive health insurance policy? This research contributes to the literature on the role of trust in the health insurer in the functioning of the healthcare system.

## Hypotheses

**H1: Enrollees have more trust in their own health insurer than in others.** Enrollees who trust their health insurer believe that the health insurer will take care of their interests [13]. When enrollees believe that another health insurer will take better care of their interests, it provides an incentive to switch to that other health insurer. When this line of reasoning is followed, enrollees are insured with the health insurer who they believe looks after their interests the most, that is to say, in which they have the most trust. We, therefore, expect that enrollees have more trust in their own health insurer than in others.

**H2: Enrollees who have more trust in health insurers in general are more likely to switch health insurers.** Trust is closely related to risk, which is derived from uncertainty about the trustee's motives, intentions, and future behaviour [8, 11]. Switching to another health insurer carries a certain risk. The lack of experience with another health insurer means there is uncertainty about whether that insurer will act, if necessary, according to their enrollees' best interests. This entails a certain risk that enrollees may be worse off with another health insurer than with their current one. Trust is shown to reduce the perceived risk [8, 10, 12]. For this reason, we expect that enrollees who have more trust in health insurers experience less of a sense of risk in switching to another health insurer. At the same time, we expect that enrollees with low trust in health insurers in general have a higher perceived risk of switching health insurers. This is because the sense of risk of poorer conditions elsewhere after switching is not eliminated by trust.

**H3: Enrollees with more trust in health insurers are more likely to opt for a policy with restrictive conditions.** Trust is expected to play a role in the willingness of enrollees to allow

their health insurer to limit their choice of providers through selective contracting [8, 16–19]. Most enrollees do not know what care they will need in the future and they do not have experience with all the care providers with whom their health insurer has concluded a contract. The risk derived from this uncertainty leads to the situation that enrollees need to have trust that the health insurer will put the enrollees' interests first when purchasing care. As trust is shown to reduce the perceived risk [8, 10, 12], we expect that trust in the health insurer plays an important role in the acceptance of selective contracting. We expect that enrollees with more trust in health insurers are more likely to opt for a policy with restrictive conditions.

## Materials and methods

### Setting

Data were collected using the Dutch Health Care Consumer Panel, an access panel managed by Nivel (the Netherlands Institute for Health Services Research) [20]. The panel collects opinions, knowledge, and experience of Dutch healthcare from citizens. The Consumer Panel is a so-called access panel. An access panel consists of a large number of people who have agreed to answer questions on a regular basis. In general, each individual panel member receives a questionnaire three to four times a year. At the time of the study in February 2022, the panel had approximately 11,500 members from the general Dutch population aged 18 and over. Their background characteristics, such as age, gender, level of education, and self-reported health status, were recorded at the start of their membership. Members can only join by invitation. Signing up on own initiative is not possible.

### Questionnaire

The questionnaire was developed by the authors who have expertise in this research topic and in conducting research based on surveys. In addition to questions about other health insurance related topics, the questionnaire asked about respondents choices in health insurance, and questions about trust in health insurers. The draft version of the questionnaire was submitted to the programme committee of the Nivel Dutch Health Care Consumer Panel who had the opportunity to give feedback. This committee consists of representatives from various interest groups in the healthcare sector, including the Ministry of Health, Welfare and Sport (VWS), the Dutch Consumers Association, and 'Zorgverzekeraars Nederland', the umbrella organisation of health insurers. As a result of their feedback, some slight changes were made to the questionnaire in order to make questions more easily understood by respondents.

### Data collection

In February 2022, the questionnaire was sent to a sample of 2,000 panel members, representative of the adult population in the Netherlands with regard to age and sex. The questionnaire could be filled in online or by post depending on the personal preference of the panel members. The panel members could fill in the questionnaire from the 10[th] of February until the 9[th] of March, 2022. Completing the questionnaire took respondents approximately 15 to 20 minutes. Two reminders were sent to respondents who had not yet completed the online questionnaire and one reminder to those who had not yet completed the paper questionnaire. In the end the questionnaire was completed by 1,125 panel members, a response rate of 56%.

### Ethical considerations

Data were assessed and processed, pseudonymised, in accordance with the panel's privacy policy, which corresponds to the General Data Protection Regulation (GDPR). Under Dutch law,

approval of a medical ethics committee is not required to conduct research with the panel [20]. Informed consent is obtained from the participants to the panel as follows: an intended participant receives an invitation to participate in the panel, including information about the purpose, scope, method and use of the panel. Based on that information, a participant can give permission to participate in the panel. This is a written consent, that since 2020 can also be given digitally. Participants are asked to complete a questionnaire a few times a year. Prior to each survey the participants receive information on the subject and the length of the survey. Participation is always voluntary and completing the questionnaire is considered consent to participate. Members of the Dutch Health Care Consumer Panel can stop their membership at any time without giving reasons.

## Measures

**Trust in the health insurer.** Trust in health insurers was measured by the Health Insurer Trust Scale (HITS), a validated scale to measure enrollees' trust developed by Zheng et al. [21]. According to Zheng et al. this scale is based on a conceptual model that assumes that insurer trust has four components that reflect overlapping aspects of insurance organisations. These are, firstly, fidelity, caring for the subject's interests or welfare. Second is competence, that is making correct decisions and avoiding mistakes. Thirdly is honesty, telling the truth and avoiding intentional falsehoods. And, finally, confidentiality, that is the proper use of sensitive information [21]. The scale consists of 11 items, in total. We used the validated Dutch translation of this scale of Hendriks et al. for this study [13].

**Trust in health insurers in general.** Firstly, we measured to what extent respondents agree with a statement about health insurers in general. This concerned only item 11 of the HITS, which, according to previous research, had the highest factor loadings and highest correlation with the scale and, therefore, can be asked individually [13]. The item is: "All in all, you trust health insurers completely". The response categories for this item were: completely agree (score 5), agree (score 4), neutral (score 3), disagree (score 2), and completely disagree (score 1). A higher score indicates more trust.

**Trust in one's own health insurer and in other health insurers.** Then all the 11 items of the HITS were asked with respect to enrollees' own insurer (see Table 1). Subsequently, this set of 11 items was repeated measuring enrollees' trust in other health insurers. For every item of the scale the words 'your health insurer' have been replaced with 'other health insurers'. The response categories for every item are completely agree (score 5), agree (score 4), neutral (score 3), disagree (score 2), and completely disagree (score 1). The scoring is reversed in the case of a negative item (items 2, 4–7, and 9). Trust is measured by the sum of the 11 item scores ranging from 11 to 55. A higher score indicates more trust. A score has only been calculated once the respondents have given an answer to all the items on the scale. The scores of respondents who did not complete the whole scale (n = 27 for own insurer and n = 28 for other insurer) or not at all (n = 86 for own insurer and n = 115 for other insurers) were converted to missing scores after checking that they were not of a specific group of respondents. This meant that their responses were not included in the analyses. To measure the internal consistency of the three items in these measures, Cronbach's alpha was used. Cronbach's alpha of the items related to the policyholders' own health insurer was 0.86. Those related to other health insurers was 0.87. Both measures, therefore, had very good reliability [22].

Three questions specifically about enrollees' trust in the purchasing strategy of health insurers were asked in addition to the validated scale developed by Zheng et al (see Table 2). These questions were derived from a study of Bes et al. [8]. Firstly, the three questions were asked with respect to the enrollees' own insurer. Then they were repeated for the enrollees' trust in

**Table 1. Health Insurer Trust Scale (HITS) (Zheng et al., 2002).**

|     | Items | Response categories |
| --- | --- | --- |
| 1. | You think the people at your health insurer are completely honest. | completely agree (score 5); agree (score 4); neutral (score 3); disagree (score 2); completely disagree (score 1) |
| 2. | Your health insurer cares more about saving money than about getting you the treatment you need. | completely agree (score 1); agree (score 2); neutral (score 3); disagree (score 4); completely disagree (score 5) |
| 3. | As far as you know, the people at your health insurer are very good at what they do. | completely agree (score 5); agree (score 4); neutral (score 3); disagree (score 2); completely disagree (score 1) |
| 4. | If someone at your health insurer made a serious mistake, you think they would try to hide it. | completely agree (score 1); agree (score 2); neutral (score 3); disagree (score 4); completely disagree (score 5) |
| 5. | You feel like you have to double check everything your health insurer does. | completely agree (score 1); agree (score 2); neutral (score 3); disagree (score 4); completely disagree (score 5) |
| 6. | You worry that private information your health insurer has about you could be used against you. | completely agree (score 1); agree (score 2); neutral (score 3); disagree (score 4); completely disagree (score 5) |
| 7. | You worry there are a lot of loopholes in what your health insurer covers that you don't know about. | completely agree (score 1); agree (score 2); neutral (score 3); disagree (score 4); completely disagree (score 5) |
| 8. | You believe your health insurer will pay for everything it is supposed to, even really expensive treatments. | completely agree (score 5); agree (score 4); neutral (score 3); disagree (score 2); completely disagree (score 1) |
| 9. | If you got really sick, you are afraid your health insurer might try to stop covering you altogether. | completely agree (score 1); agree (score 2); neutral (score 3); disagree (score 4); completely disagree (score 5) |
| 10. | If you have a question, you think your health insurer will give a straight answer. | completely agree (score 5); agree (score 4); neutral (score 3); disagree (score 2); completely disagree (score 1) |
| 11. | All in all, you have complete trust in your health insurer. | completely agree (score 5); agree (score 4); neutral (score 3); disagree (score 2); completely disagree (score 1) |

other health insurers. In the second case, the words "your health insurer" were replaced by "other health insurer" for all three items. The response categories for each item are the same as with the Health Insurance Trust Scale, being: completely agree (score 5), agree (score 4), neutral (score 3), disagree (score 2), and completely disagree (score 1). Trust in the purchasing strategy of health insurers is measured by the sum of the three item scores, ranging from 3 to 15. A higher score indicates more trust. A score is only calculated if respondents answered all three questions. The scores of respondents who did not complete the whole the scale (n = 16 for own insurer and n = 14 for other insurer) or not at all (n = 91 for own insurer and n = 120 for other insurers) were converted to missing scores after checking that they were not of a

**Table 2. Measure of enrollees' trust in health insurers' purchasing strategy.**

|     | Items | Response categories |
| --- | --- | --- |
| 1. | I trust my health insurer/other health insurers to choose the best care providers. | completely agree (score 5); agree (score 4); neutral (score 3); disagree (score 2); completely disagree (score 1) |
| 2. | I trust my health insurer/other health insurers not to compromise on quality in order to keep the price down. | completely agree (score 5); agree (score 4); neutral (score 3); disagree (score 2); completely disagree (score 1) |
| 3. | I trust my health insurer/other health insurers to choose the best care for me at the best price. | completely agree (score 5); agree (score 4); neutral (score 3); disagree (score 2); completely disagree (score 1) |

specific group of respondents. This meant that their responses were not included in the analyses. Cronbach's alpha was used to measure the internal consistency of the three items in these measures. The figure for Cronbach's alpha of both the items related to the policyholders' own health insurer, and the items related to other health insurers, was 0.89. Both measures, therefore, had very good reliability [22].

**Enrollees' switching behaviour.** We looked at whether enrollees switched health insurers as of 2022. The question that has been asked to measure enrollees' switching behaviour in that year was: 'Have you switched health insurers as of 2022?' The response categories on this question were: (1) 'No, but I have considered switching health insurers'; (2) 'No, nor have I considered switching health insurers'; (3) 'Yes, but only for basic insurance'; (4) 'Yes, but only for supplemental insurance'; and (5) 'Yes, both for basic and supplemental insurance'. Since the interest was whether they actually switched or not, regardless of basic or supplemental insurance, a dummy variable was created (1 = Yes, 0 = No).

**Choosing a restrictive health insurance policy.** With regard to the choice of a restrictive health insurance policy, we specifically focused on four health insurance policies in the Netherlands with restrictive conditions in 2022 which offer a choice of fewer hospitals. These four policies are: Pro Life Principe Polis Budget, Gewoon ZEKUR Polis, ZieZo Selectief and Zilveren Kruis Basis Budget. To measure this enrolment, we asked respondents: Do you have one of the following health insurance policies: Pro Life Principe Polis Budget, Gewoon ZEKUR Polis, ZieZo Selectief or Zilveren Kruis Basis Budget? The response categories on this question were: (1) 'No', (2) 'Yes', (3) 'I do not know'. Since the interest was not so much about which policy people have, but whether they have a policy with restrictive conditions, a dummy variable was created (1 = Yes, 0 = No). The answer option, 'I don't know', has been converted into a missing value (n = 116).

**Background variables.** The background variables that are known from the panel members and are included concern: age (continuous): gender (0 = male, 1 = female); educational level (1 = low—none, primary school or pre- vocational education, 2 = middle—secondary or vocational education, 3 = high—professional higher education or university); and self-reported health status, which was measured using an item derived from the 36-Item Short Form Survey (SF-36) (1 = bad/fair, 2 = good, 3 = very good/excellent).

## Data analysis

Descriptive statistics were computed to describe the characteristics of the study population and to analyse the level of enrollees' trust in health insurers in general. Because the survey respondents were not fully representative of all Dutch citizens aged 18 years and older in terms of the combination of age and gender, we applied weighting factors by age and gender to the descriptive statistics related to enrollees' trust in health insurers to correct for this. These weighting factors were calculated by dividing the amount of males and females per age group (18–49, 50–64, and 65+) in the study sample with the corresponding age groups of the Dutch general population as provided by Statistics Netherlands (CBS) [23]. The weighting factors ranged from 0.80 to 1.67 in males, and 0.82 to 1.75 in females.

In order to test hypothesis 1, paired t-tests were conducted to examine the difference between trust in one's own insurer and trust in other insurers. We examined this for trust in the health insurer in general, as well as specifically in their purchasing strategy.

In order to test hypothesis 2, a multivariate logistic regression model was applied. Here the dependent variable was the answer to the question about whether enrollees have switched health insurers as of 2022 (yes/no). General trust in health insurers (Item 11 HITS) was applied as the independent variable of interest (model 1). Other background variables included in the model were gender, age, the level of education, and self-reported health status.

For testing hypothesis 3, three multivariate logistic regression models were applied. Here the dependent variable was the answer to the question whether enrollees have a health insurance policy with restrictive conditions (yes/no). The independent variables of interest were: trust in health insurers in general (Item 11 HITS) (model 2); trust in enrollees' own health insurer (HITS) (model 3), and; trust in the purchasing strategy of enrollees' own health insurer (model 4). Once again, the background variables included in all three models were gender, age, the level of education, and self-reported health status.

A significance level of 5% (p≤0.05) was maintained for the analyses. All analyses were performed using STATA version 16.1.

## Results

### Descriptive statistics

Half (50%) of the respondents were women. Respondents were, on average, 57 years old (Table 3). Furthermore, 47% of the respondents were highly educated, and 19% rated their health as bad or fair. The average score for trust in their own health insurer, measured by the

**Table 3. Descriptive statistics of the respondents.**

| | Number of respondents (n) | Percentage (%) or mean (SD) |
|---|---|---|
| **Gender** | **1125** | |
| Male | 566 | 50% |
| Female | 559 | 50% |
| **Age** | **1125** | 57 (15.97) |
| 18–39 years | 227 | 20% |
| 40–64 years | 555 | 49% |
| 65 years and older | 343 | 30% |
| **Education** | **1098** | |
| Low (none, primary school or pre-vocational education) | 109 | 10% |
| Middle (secondary or vocational education) | 475 | 43% |
| High (professional higher education or university) | 514 | 47% |
| **Health (self-reported)** | **989** | |
| Bad/fair | 187 | 19% |
| Good | 473 | 48% |
| Very good/excellent | 329 | 33% |
| **Trust in own health insurer (HITS)** | 1012 | 36.73 (6.05) (range 11–55) |
| **Trust in other health insurers (HITS)** | 982 | 33.93 (5.17) (range 11–54) |
| **Trust in one's own health insurer's purchasing strategy** | 1018 | 9.75 (2.46) (range 3–15) |
| **Trust in other health insurers' purchasing strategy** | 991 | 9.04 (1.97) (range 3–15) |
| **Have you switched health insurers as of 2022?** | **1086** | |
| Yes | 105 | 10% |
| No | 981 | 90% |
| **Do you have one of the following health insurance policies: Pro Life Principe Polis Budget, Gewoon ZEKUR Polis, ZieZo Selectief, or Zilveren Kruis Basis Budget?** | **960** | |
| Yes | 106 | 11% |
| No | 854 | 89% |

**Table 4. Descriptive statistics to examine enrollees' trust in health insurers in general.**

| | unweighted | | weighted[a] | |
|---|---|---|---|---|
| | Obs | % | Obs | % |
| **Trust in health insurers in general: All in all, you trust health insurers completely (Item 11 HITS)** | **1040** | | **1036.42** | |
| completely agree | 51 | 5% | 51.28 | 5% |
| agree | 317 | 30% | 309.13 | 30% |
| neutral | 453 | 44% | 442.20 | 43% |
| disagree | 151 | 15% | 159.65 | 15% |
| completely disagree | 68 | 7% | 74.16 | 7% |

a. Results corrected with weighting factors by age and gender.

HITS, was 36.73, and the average trust score in other health insurers, 33.93. Corrected with weighting factors by age and gender, these scores are 36.62 and 33.99, respectively (not in table). The average total score on the 3-items measure, with regard to the health insurers' purchasing strategy, was 9.75 for one's own insurer, and 9.04 for other health insurers. When weighting factors by age and gender are applied, these scores are 9.64 and 9.02, respectively (not in table). Ten percent of the respondents indicated that they have switched health insurers as of 2022. Furthermore, 11% indicated that they have one of the four health insurance policies with restrictive conditions in the Netherlands which offer a choice of fewer hospitals.

## What is the level of enrollees' trust in health insurers, and does it differ between their own and other health insurers?

The results show that 35% of the respondents indicated that they agree, or completely agree, with the statement that they trust health insurers completely (Table 4). Furthermore, 22% of the respondents disagree, or completely disagree, with the statement, while 44% of the respondents indicated they are neutral. Corrected with weighting factors by age and gender, these percentages are almost identical (Table 4).

The mean score of the respondents in the paired t-test on the HITS with regard to trust in their own health insurer is 36.67 out of 55 (Table 5). This score is on average 2.75 higher than mean score on the HITS with regard to trust in other health insurers, which is 33.91 out of 55. This difference is statistically significant (p<0.001).

The mean score of the respondents in the paired t-test on the HITS with regard to trust in their own health insurer's purchasing strategy is 9.76 out of 15 (Table 6). This score is on average 0.71 higher than mean score on the HITS with regard to trust in other insurers' purchasing strategy, which is 9.05 out of 15. This difference is statistically significant (p<0.001). The results of both paired t-tests are in line with hypothesis 1.

**Table 5. Paired t-test to examine the difference between trust in one's own insurer and trust in other insurers.**

| | Obs | Mean | SE | 95% Conf. Interval | | Ha: mean(diff)! = 0 |
|---|---|---|---|---|---|---|
| **HITS: Trust in one's own health insurer** | 962 | 36.67 | 0.19 | 36.29 | 37.05 | |
| **HITS: Trust in other health insurers** | 962 | 33.91 | 0.17 | 33.59 | 34.24 | |
| **Difference** | 962 | 2.75 | 0.13 | 2.50 | 3.01 | p<0.001* |

* Significant p-value

**Table 6. Paired t-test to examine the difference between trust in one's own insurer's purchasing strategy and trust in other insurers' purchasing strategy.**

|  | Obs | Mean | SE | 95% Conf. Interval | | Ha: mean(diff)! = 0 |
|---|---|---|---|---|---|---|
| Trust in one's own health insurer's purchasing strategy | 975 | 9.76 | 0.08 | 9.60 | 9.91 | |
| Trust in other health insurers' purchasing strategy | 975 | 9.05 | 0.06 | 8.93 | 9.17 | |
| Difference | 975 | 0.71 | 0.06 | 0.60 | 0.82 | p<0.001* |

* Significant p-value

## What is the relationship between trust in health insurers and enrollees' switching behaviour?

Model 1 in Table 7 shows no significant relationship between general trust in health insurers and whether enrollees have switched health insurers as of 2022 (OR = 0.83, p = 0.12). This result is not in line with hypothesis 2. Hypothesis 2 is, therefore, not accepted. If we look at the background characteristics, older enrollees have switched health insurers as of 2022 slightly less often than younger enrollees (OR = 0.94, p<0.001).

## What is the relationship between trust in health insurers and enrollees' choice of a restrictive health insurance policy?

In models 2–4, presented in Table 8, no significant associations were found between trust in health insurers and enrollees' choice of a restrictive health insurance policy. The results show that enrollees' trust, both in general, and specifically in their health insurer and its purchasing strategy, is not associated with their choice for a policy with restrictive conditions. These results are not in line with hypothesis 3. Hypothesis 3 is therefore not accepted. If we look at the background characteristics, then especially men and people with a low level of education are more likely to have a health insurance policy with restrictive conditions.

**Table 7. Multivariate logistic regression to examine the associations between switching behaviour and general trust in health insurers.**

|  |  | Model 1 (n = 944) | |
|---|---|---|---|
|  |  | Y: Have you switched health insurers as of 2022? (1 = Yes, 0 = No) | |
|  |  | Odds Ratio | P-value |
| General trust in health insurers (Item 11 HITS)[b] |  | 0.83 | 0.12 |
| Gender | Male | Reference | |
|  | Female | 1.25 | 0.33 |
| Age[b] |  | 0.94 | <0.001* |
| Education[c] | Low | Reference | |
|  | Middle | 1.57 | 0.47 |
|  | High | 2.04 | 0.25 |
| Health (self-reported) | Bad/fair | Reference | |
|  | Good | 1.11 | 0.77 |
|  | Very good/excellent | 0.89 | 0.74 |
| Constant |  | 0.04 | <0.001* |

* Significant p-value

b. Centred around the mean

c. Low = none, primary school or pre-vocational education. Middle = secondary or vocational education.

High = professional higher education or university.

**Table 8. Multivariate logistic regression to examine the associations between having a health insurance policy with restrictive conditions and different types of trust in the health insurer.**

Y: Do you have one of the following health insurance policies: Pro Life Principe Polis Budget, Gewoon ZEKUR Polis, ZieZo Selectief or Zilveren Kruis Basis Budget? (1 = Yes, 0 = No)

| Model 2 (n = 847) Examining the association with general trust in health insurers | | Odds Ratio | P-value | Model 3 (n = 835) Examining the association with trust in one's own health insurer | | Odds Ratio | P-value | Model 4 (n = 837) Examining the association with trust in one's own health insurer's purchasing strategy | | Odds Ratio | P-value |
|---|---|---|---|---|---|---|---|---|---|---|---|
| General trust in health insurers (item 11 HITS, range 1–5)[b] | | 1.06 | 0.66 | Trust in enrollees' health insurer (HITS, range 11–55)[b] | | 0.99 | 0.69 | Trust in purchasing strategy of enrollees' health insurer (range 3–15)[b] | | 0.98 | 0.73 |
| Gender | Male | Ref. | | Gender | Male | Ref. | | Gender | Male | Ref. | |
| | Female | 0.63 | 0.05* | | Female | 0.63 | 0.05* | | Female | 0.64 | 0.06 |
| Age[b] | | 1.02 | 0.02* | Age[b] | | 1.02 | <0.01* | Age[b] | | 1.02 | <0.01* |
| Education[c] | Low | Ref. | | Education[c] | Low | Ref. | | Education[c] | Low | Ref. | |
| | Middle | 0.45 | 0.02* | | Middle | 0.42 | <0.01* | | Middle | 0.42 | 0.01* |
| | High | 0.32 | <0.01* | | High | 0.29 | <0.001* | | High | 0.29 | <0.001* |
| Health (self-reported) | Bad/fair | Ref. | | Health (self-reported) | Bad/fair | Ref. | | Health (self-reported) | Bad/fair | Ref. | |
| | Good | 1.14 | 0.68 | | Good | 1.07 | 0.84 | | Good | 1.08 | 0.80 |
| | Very good/ excellent | 1.53 | 0.23 | | Very good/ excellent | 1.43 | 0.31 | | Very good/ excellent | 1.47 | 0.27 |
| Constant | | 0.27 | <0.01* | Constant | | 0.31 | <0.01* | Constant | | | <0.01* |

\* Significant p-value

b. Centred around the mean

c. Low = none, primary school or pre-vocational education. Middle = secondary or vocational education. High = professional higher education or university.

## Discussion

In this study, we investigated how enrollees' trust in health insurers is associated with their choice of a health insurance policy in the Netherlands. The results showed, first of all, that 35% of the respondents indicated that they agree, or completely agree, with the statement that they trust health insurers completely. In addition, we found that the level of enrollees' trust is slightly, but significantly, different between their own and other health insurers. Enrollees have more trust in their own health insurer than in other health insurers. Lastly, against our expectation, we found no significant associations between trust in health insurers and whether enrollees have switched health insurers or have chosen a policy with restrictive conditions.

The level of enrollees' trust found in health insurers (35%) is much lower than the known level of trust found in 2020 in general practitioners (89%), medical specialists (91%) or hospitals (83%) in the Netherlands [24]. In addition, it also appears to be lower than that which literature suggests is trust in health insurers in Switzerland and Germany, which have similar healthcare systems to the Netherlands [25–28]. Of the respondents, 22% indicated that they distrusted health insurers. Maarse et al. (2019) suggest several reasons why people may distrust health insurers. For example it might be due to a lack of health insurance literacy, the fact that health insurers are seen as for-profit organisations, the belief that third parties should not interfere in the doctor-patient relationship, and the media influencing public opinion about health insurers [25]. However, literature on reasons why trust in health insurers is low is scarce. Further research on this topic may provide more insight into this.

Furthermore, we found that the level of enrollees' trust is slightly different between their own and other health insurers. This applies both to trust in general and specifically to trust in the purchasing strategy. In line with our hypothesis, enrollees are thus insured with the health insurer that they believe looks after their interests the most–that is in which they have the most trust. This may be because positive experiences, or the absence of negative experiences, resulted in enrollees feeling satisfied, as earlier research has shown that satisfaction with the health insurer was a strong predictor of trust in them [29]. In addition, we know from literature that disputes with the insurer in the past are related to a lower trust in the health insurer [30].

In addition, this study showed that general trust in health insurers is not related to whether enrollees have switched health insurers as of 2022. This was not as expected. A possible explanation could be the fact that our data shows that young enrollees have switched slightly more often as of 2022. Also, literature shows that people who switch are relatively younger and healthier [31, 32]. For these groups, trust might not play such an important role because they might not use much care anyway and therefore do not expect to have to rely on their health insurer. Instead, the level of premium might be the most important reason to switch, as is confirmed in other literature [33–35].

The absence of a relationship between trust in health insurers and enrollees' switching behaviour could also be due to the type of trust we used to analyse this relationship. We could only examine the association between *general trust* in health insurers and whether respondents switched health insurers. We might have found different results if it was possible to examine the association between the level of trust in one's own health insurer and the choice to switch or not. Even though we have measured trust in one's own health insurer, it was not possible to use this measurement to examine its relationship with switching as of 2022. Our study was conducted in February, after the period in which switching health insurers can take place instead of before. Trust in one's own health insurer was, therefore, related to their new health insurer for respondents who had switched as of 2022. Future research could focus on whether the level of trust in enrollees' own health insurer, preferably compared with the level of trust in other health insurers, affects enrollees' switching behaviour.

Finally, this study showed that trust, both in general, and specifically in their health insurer and its purchasing strategy, is not related to the choice of a policy with restrictive conditions. This finding seems inconsistent with previous research by Bes et al. (2013), which found that trust in the health insurer was found to be an important condition for the acceptance of selective contracting among their enrollees [8]. This possible contradiction could be caused by a lack of knowledge amongst enrollees about policies with restrictive conditions. Indeed, previous research has shown that the awareness of the restrictive conditions of this type of health insurance policies is low [36]. According to our hypothesis, more trust in the health insurer is essential in order to choose such a policy, as trust is important for the acceptance of selective contracting. However, if enrollees do not know that selective contracting is part of such health insurance policies, the role of trust in choosing a health insurance policy may be negated. Further research could focus on the relationship between knowledge about policy conditions and the extent to which trust plays a role in choosing a health insurance policy.

For the functioning of the current healthcare system based on managed competition as intended, it is important that enough enrollees change health insurers each year, or consciously consider doing so, in order that insurers feel that there is a genuine threat of enrollees leaving them. Furthermore, it is important that health insurers are able to guide enrollees to preferred providers via selective contracting. This is because the more enrollees choose restrictive health insurance policies, the more selective contracting may become an effective key element in creating competition between healthcare providers. The results of this study show that low general trust in insurers does not necessarily pose a threat to the functioning of the

healthcare system in regard to switching insurers. General trust seems not to be related to switching behaviour and the choice of a policy with restrictive conditions. However, this does not mean that it is not important to maintain, or increase, the level of trust in health insurers because trust might also be important for other elements related to the functioning of the healthcare system based on managed competition. For example, according to Maarse and Jeurissen, low institutional trust in health insurers may cause inefficiency in the health insurance market, undermine the legitimacy of health insurance, and eventually weaken solidarity in health insurance [25, 37]. In addition, literature suggests that with lack of trust in the health insurer, people do not choose policies with selective contracting and do not want to be channeled to preferred providers [16, 38]. Furthermore, low trust is negatively related with enrollees 'willingness to receive healthcare advice from the health insurer [39]. Both, policies with restrictive conditions and healthcare advice, are instruments that may give health insurers the ability to channel enrollees to preferred providers. The ability to channel enrollees to preferred providers is essential for health insurers to act as prudent purchasers of care [16, 40–42]. An inability to use these instruments to channel enrollees could lead to a weakening of the bargaining position of health insurers towards health care providers. This may result in health insurers being restrained to steer on cost and/or quality of care. Further research may identify which other elements in the functioning of the healthcare system based on managed competition are affected by enrollees' trust in their health insurer. This could offer more insight into the degree of importance of increasing trust in the health insurer. Accordingly, future research should focus the possibilities for increasing trust in health insurers, as there is a lack of literature on this topic.

## Strengths and limitations

A strength of the methodology applied in this study is that trust in health insurers was measured using a validated scale (HITS). The use of this scale contributes to the validity of this study. Another strength is the high number of respondents (n = 1,125), which increases the reliability of the study. Furthermore, by sending questionnaires both by mail and online meant that even people with low digital skills could participate in the survey. However, the respondents were not fully representative of the Dutch population aged 18 years and older. Our respondents are relatively older and more educated thus underrepresenting the younger and less educated [23]. This could affect the level of trust we measured, as several studies have found a relationship between age or level of education with trust [9, 43–45]. However, the constraint that respondents are relatively older is addressed by the fact that we corrected the descriptive statistics regarding enrollees' trust in health insurers with weighting factors by age and gender. Furthermore, our regression results are not expected to have been affected, as all subgroups are sufficiently large to perform association analyses.

Another limitation of this study is related to the relationship between trust in the health insurer and whether or not enrollees choose a restrictive health insurance policy. Whether respondents have a policy with restrictive conditions is self-indicated. We have to assume that respondents are aware of the name of their health insurance policy, and therefore filled in the question correctly. Furthermore, one health insurance policy provided by a really small insurance concern with less than 1% of the market share was not included in our question, but this should hardly have affected our measurement given its market share. A comparison of our data with the literature shows that 11% of our respondents reported having a policy with restrictive conditions, which is lower than the national rate of 21.8% reported in the literature [46]. Nevertheless, the proportion of young people between aged 18 and 40 years old is underrepresented among our respondents (20% compared to 34% of the national Dutch adult

population), while literature shows that most enrollees opting for such a policy are of this age group [47]. Thus, a deviation between these percentages does not, by definition, mean that there is an insufficient measurement. Furthermore, by presenting information in advance of the question about the existence of different types of policies, and by adding the answer option 'I do not know', we expect that incorrect completion of this question was minimised.

## Conclusions

This study showed that enrollees' trust in health insurance in the Netherlands is relatively low and that trust in their own insurer is slightly higher than trust in other insurers. Furthermore, this study did not find a relationship between trust in the health insurer and whether enrollees have switched health insurers. Lastly, no relationship was found between enrollees' trust in the health insurer and choosing a policy with restrictive conditions. This study contributes to the limited research on the field of trust in health insurers. The results of this study do not indicate that low trust in health insurers poses a threat to the functioning of the healthcare system with regard to switching behaviour and the choice of a policy with restrictive conditions. Nevertheless, it may still be important to give attention to increasing trust in health insurers, because trust can be necessary for other elements of the functioning of the healthcare system. For example, according to literature the level of trust seems to be related to the ability to channel enrollees to preferred providers, which is essential for the execution of the health insurers' role as prudent purchasers of care. Further research may identify which other elements in the functioning of the healthcare system based on managed competition are affected by enrollees' trust in their health insurer. In addition, as this study found that the level of enrollees' trust in health insurers is relatively low, future research could focus on possibilities for increasing trust in health insurers.

## Acknowledgments

We would like to thank the members of the Dutch Health Care Consumer Panel who filled out the questionnaire.

## Author Contributions

**Supervision:** Judith D. de Jong.

**Writing – original draft:** Frank J. P. van der Hulst.

**Writing – review & editing:** Anne E. M. Brabers, Judith D. de Jong.

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
