## [Decision Letter · Decision Letter 0]

25 Apr 2023

PONE-D-23-04367What role does enrollees’ trust in health insurers play when choosing health insurance?PLOS ONE

Dear Dr. Hulst,

Thank you for submitting your manuscript to PLOS ONE. After careful consideration, we feel that it has merit but does not fully meet PLOS ONE’s publication criteria as it currently stands. Therefore, we invite you to submit a revised version of the manuscript that addresses the points raised during the review process.

We look forward to receiving your revised manuscript.

Kind regards,

Tai Ming Wut

Academic Editor

PLOS ONE

Journal Requirements:

"The data collection of this study was funded by the Dutch Ministry of Health, Welfare and Sport. The funder had no role in the design, execution and writing of the study."

"The data collection of this study was funded by the Dutch Ministry of Health, Welfare and Sport. The funder had no role in the design, execution and writing of the study.."

"The data collection of this study was funded by the Dutch Ministry of Health, Welfare and Sport. The funder had no role in the design, execution and writing of the study."

Reviewers' comments:

Reviewer's Responses to Questions

**Comments to the Author**

1. Is the manuscript technically sound, and do the data support the conclusions?

Reviewer #1: Partly

2. Has the statistical analysis been performed appropriately and rigorously? 

Reviewer #1: No

3. Have the authors made all data underlying the findings in their manuscript fully available?

Reviewer #1: No

4. Is the manuscript presented in an intelligible fashion and written in standard English?

Reviewer #1: Yes

5. Review Comments to the Author

Reviewer #1: My largest concern is the panel that responded to the survey seems to be substantially different than the general population. The authors acknowledge this as a limitation from lines 400-408. I would like to know the following things:

1) Is the response rate for this survey similar to other surveys administered to this panel?

2) Is it possible to validate the % of people in restrictive plans with public data. I am unfamiliar with the details of the Dutch health insurance systems but I would imagine market share information should be available.

3) How different is table 3 descriptive statistics for the responding group to the non-responding or general panel group?

I fear that approaches which weigh the data to match population observed characteristics, especially on income and education will still be missing a substantial unobserved characteristic of bureaucratic navigation skill and education. With a voluntary sample that is substantially different from the general panel, I am not sure how to generalize or interpret these results.

Other comments:

a) I like how Table 6 is presented.

b) Switching behaviors --- what is the look back period for the switch insurance? I don't know if it is in the past contract period or ever.

c) Table 7 -- I am unsure how to interpret AGE -- do older people have higher or lower odds of switching? I can't tell direction here. Please clarify.

d) P=0.00 makes no sense to me, is it better to say p<.01 or p<.0001 as there is some miniscule chance of a random draw from a distribution centered on the null to produce a result as unusual as the ones that are observed.

e) Table 8 hints that there is a rational response to more restrictive insurance by individuals who are less likely to use insurance or who are severely cash-constrained. Is it possible to incorporate income into these anaylses to further examine the role of resource availability and choice?

6. PLOS authors have the option to publish the peer review history of their article (what does this mean?). If published, this will include your full peer review and any attached files.

Reviewer #1: No

---

## [Author Response · Author response to Decision Letter 0]

26 Jun 2023

Comments from the academic editor:

Comment #1: Please ensure that your manuscript meets PLOS ONE's style requirements, including those for file naming. The PLOS ONE style templates can be found at 

Reaction to comment #1: 

Thank you for addressing PLOS ONE’s style requirements. We have amended the heading levels, table titles, reference citations, acknowledgments section, and the title/authors/affiliations page of the manuscript. 

Comment #2: We note that the grant information you provided in the ‘Funding Information’ and ‘Financial Disclosure’ sections do not match. 

Reaction to comment #2: 

A grant number does not apply to this study. However, the information provided in the ‘Funding Information’ and ‘Financial Disclosure’ section need to match. We therefore amended the information provided in the ‘Funding Information’ by adding the Ministry of Health, Welfare and Sport as Funding Source.

Comment #3: Thank you for stating the following financial disclosure: 

"The data collection of this study was funded by the Dutch Ministry of Health, Welfare and Sport. The funder had no role in the design, execution and writing of the study."

Reaction to comment #3: 

We have amended the Role of Funder statement and added it to our cover letter. Thank you for changing the online submission form on our behalf. The Role of Funder statement has been amended into: 

“The data collection of this study was funded by the Dutch Ministry of Health, Welfare and Sport. The funders had no role in study design, data collection and analysis, decision to publish, or preparation of the manuscript.” 

Comment #4: Thank you for stating the following in the Acknowledgments Section of your manuscript: 

"The data collection of this study was funded by the Dutch Ministry of Health, Welfare and Sport. The funder had no role in the design, execution and writing of the study.."

"The data collection of this study was funded by the Dutch Ministry of Health, Welfare and Sport. The funder had no role in the design, execution and writing of the study."

Reaction to comment #4: 

Thank you for addressing that the funding information should not appear in the Acknowledgements sections or other areas of our manuscript. We have removed it now. As mentioned in the rection to comment #3, we have added the Role of Funder statement to our cover letter, and have amended it into:

“The data collection of this study was funded by the Dutch Ministry of Health, Welfare and Sport. The funders had no role in study design, data collection and analysis, decision to publish, or preparation of the manuscript.” 

Comment #5: We note that you have indicated that data from this study are available upon request. PLOS only allows data to be available upon request if there are legal or ethical restrictions on sharing data publicly. For more information on unacceptable data access restrictions, please see http://journals.plos.org/plosone/s/data-availability#loc-unacceptable-data-access-restrictions. 

Reaction to comment #5: 

We have did now upload the minimal anonymized data set necessary to replicate our study findings to EASY. In the revised version of our cover letter, we explained the data availability as follows:

“The minimal anonymized data set necessary to replicate our study findings are uploaded to EASY (https://doi.org/10.17026/dans-x5m-hppm). The data set may only be used under the conditions laid down in the privacy regulations of the Dutch Health Care Consumer Panel. Therefore, the dataset is available on request. Data requests will be assessed against these regulations. Additionally, the same minimal anonymized data set is available upon request from prof. Judith D. de Jong (j.dejong@nivel.nl), project leader of the Dutch Health Care Consumer Panel, or the secretary if this panel (conusmentenpanel@nivel.nl). The Dutch Health Care Panel had a program committee, which supervises processing the data of the Dutch Health Care Consumer Panel and decides about the use of the data. This program committee consists of representatives of the Dutch Ministry of Health, Welfare and Sport, the Health Care Inspectorate, Zorgverzekeraars Nederland (Association of Health Care Insurers in the Netherlands), the National Health Care Institute, the Federation of Patients and Consumer Organisations in the Netherlands, the Dutch Healthcare Authority and the Dutch Consumers Association. All research conducted within the Consumer Panel has to be approved by this program committee. The committee assesses whether a specific research fits within the aim of the Consumer Panel, which is to strengthen the position of the health care user.”

Comment #6: Your ethics statement should only appear in the Methods section of your manuscript. If your ethics statement is written in any section besides the Methods, please delete it from any other section.

Reaction to comment #6: 

We have erased the ethics statement from the declaration section (which is now the acknowledgement section) of the manuscript, so this information is now only present in the Materials and Methods section of our manuscript. 

Comments from the reviewers: 

Reviewer #1

Comment #1: My largest concern is the panel that responded to the survey seems to be substantially different than the general population. The authors acknowledge this as a limitation from lines 400-408. I would like to know the following things:

1) Is the response rate for this survey similar to other surveys administered to this panel?

2) Is it possible to validate the % of people in restrictive plans with public data. I am unfamiliar with the details of the Dutch health insurance systems but I would imagine market share information should be available.

3) How different is table 3 descriptive statistics for the responding group to the non-responding or general panel group?

I fear that approaches which weigh the data to match population observed characteristics, especially on income and education will still be missing a substantial unobserved characteristic of bureaucratic navigation skill and education. With a voluntary sample that is substantially different from the general panel, I am not sure how to generalize or interpret these results.

Reaction to comment #1: 

The reviewer’s concern is that the panel that responded to the survey might be substantially different than the general population. Below, we provide the answers to the three things the reviewer would like to know with respect to this. 

1) Yes, the response rate of 56% is similar to other surveys administered to this panel. In general, the response rate of surveys administered to this panel is between 45-60% [1-4]. Literature on Survey research suggests that good survey research has an adequate response rate of at least 40% [5]. Our response rate is higher than [6], or reasonably in line [7] with, average response rates in survey research reported in the literature.

2) As stated in lines 459-461 of the revised manuscript, a comparison of our data with the literature shows that 11% of our respondents reported having a policy with restrictive conditions, which is lower than the national rate of 21.8% reported in the literature [8]. Nevertheless, as we state in lines 461-465, the proportion of young people between aged 18 and 40 years old is underrepresented among our respondents (20% compared to 34% of the national Dutch adult population). As literature shows that most enrollees opting for such a policy are of this age group [9], a deviation between these percentages does not, by definition, mean that there is an insufficient measurement.

3) In the responding group the proportion of young people between aged 18 and 40 years old is underrepresented (20% compared to 34% of the national Dutch adult population) [10]. As we took a sample which is representative regarding gender and age of the adult Dutch population, this means that in the non-respons group younger adults are overrepresented. With regard to gender, the responding group is comparable with the general national Dutch adult population, as well as the total Dutch Health Care Consumer Panel (50% male, 50% female) [10, 11].

With regard to education and health-status, the sample was drawn at random from the total Dutch Health Care Consumer Panel. The health-status of responding group (81% good or very good/excellent) is comparable with the general national Dutch adult population (80,5% good or very good) [12]. As stated in lines 446-447, our respondents are more educated compared to the general national Dutch adult population, thus underrepresenting the less educated [10].

Furthermore, the reviewer indicates to fear that approaches which weigh the data to match population observed characteristics, especially on income and education will still be missing a substantial unobserved characteristic of bureaucratic navigation skill and education. With a voluntary sample that is substantially different from the general panel, the reviewer indicates that he or she is not sure how to generalize or interpret these results. 

As we state in lines 446-449, the circumstance that older and more educated enrollees are overrepresented could affect the level of trust we measured. The level of trust we measured is probably higher than actually in the general population, because there were more elderly people in the response group and they might have higher levels of trust [13, 14, 15]. Therefore, following the reviewer's comments, we decided to apply weighting factors by age and gender to the descriptive statistics related to enrollees’ trust in health insurers to correct for this. 

To the Materials and Methods section of the manuscript, we added in lines 272-279 the following text with regard to applying weighting factors: 

“Because the survey respondents were not fully representative of all Dutch citizens aged 18 years and older in terms of the combination of age and gender, we applied weighting factors by age and gender to the descriptive statistics related to enrollees’ trust in health insurers to correct for this. These weighting factors were calculated by dividing the amount of males and females per age group (18–49, 50–64, and 65+) in the study sample with the corresponding age groups of the Dutch general population as provided by Statistics Netherlands (CBS)[23]. The weighting factors ranged from 0.80 to 1.67 in males, and 0.82 to 1.75 in females.”

Furthermore, to the Results section, we added the following results in lines 302-303: 

“Corrected with weighting factors by age and gender, these scores are 36.62 and 33.99, respectively (not in table).”

In addition, in lines 305-306 of the results section, we added the following:

“When weighting factors by age and gender are applied, these scores are 9.64 and 9.02, respectively (not in table).”

Furthermore, we added a column with weighted results to Table 4. In relation to that, we added the following to the text of the Results section in lines 317-318:

“Corrected with weighting factors by age and gender, these percentages are almost identical (Table 4).” 

Lastly, we added in lines 449-451 the following text to the limitations about this:

“However, the constraint that respondents are relatively older is addressed by the fact that we corrected the results regarding enrollees' trust in health insurers in general with weighting factors by age and gender.” 

We did not apply weighting of these results with regard to income and education. We did not for income, because we do not measured respondents' income in this questionnaire, so we do not have this data. As for education, we made the choice not to correct for this with weight factors, because, apart from applying weight factors for age and gender, we did not want to tinker too much with the results. We have therefore included this point as a limitation in lines 446-449, as mentioned earlier.

Furthermore, as stated in lines 451-452 our regression results are not expected to have been affected by the circumstance that older and more educated enrollees are overrepresented, as all subgroups are sufficiently large to perform association analyses. Inclusion of the background variables in the logistic regression models, including age and education, ensures that there is control for confounders.

References:

1. Holst, L., Victoor, A., Brabers, A., Rademakers, J., & de Jong, J. (2022). Health insurance literacy in the Netherlands: The translation and validation of the United States’ Health Insurance Literacy Measure (HILM). Plos one, 17(9), e0273996.

2. van der Hulst, F. J., Brabers, A. E., & de Jong, J. D. (2023). The relation between trust and the willingness of enrollees to receive healthcare advice from their health insurer. BMC Health Services Research, 23(1), 1-11.

3. van der Hulst, F. J., Holst, L., Brabers, A. E., & de Jong, J. D. (2022). To what degree are health insurance enrollees in the Netherlands aware of the restrictive conditions attached to their policies?. Health Policy, 126(7), 693-703.X

4. Brabers, A. E., Rademakers, J. J., Groenewegen, P. P., Van Dijk, L., & De Jong, J. D. (2017). What role does health literacy play in patients' involvement in medical decision-making?. PloS one, 12(3), e0173316.

5. Story, D. A., & Tait, A. R. (2019). Survey research. Anesthesiology, 130(2), 192-202. 

6. Wu, M. J., Zhao, K., & Fils-Aime, F. (2022). Response rates of online surveys in published research: A meta-analysis. Computers in Human Behavior Reports, 100206.

7. Holtom, B., Baruch, Y., Aguinis, H., & A Ballinger, G. (2022). Survey response rates: Trends and a validity assessment framework. human relations, 00187267211070769.

8. NZa. Monitor zorgverzekeringsmarkt 2022. Utrecht: NZa; 2022 1 November 2022.

9. NZa. Polissen met beperkt aantal contracten in de medisch specialistische zorg. Analyse van aantal, aard en kenmerken van deze polissen, in opdracht van het ministerie van VWS. Utrecht: NZa; 2020 8 October 2020.

10. Bevolking op 1 januari en gemiddeld; geslacht, leeftijd en regio. In: CBS, editor. 2022.

11. Brabers A, de Jong J. Nivel Consumentenpanel Gezondheidszorg: basisrapport met informatie over het panel 2022 [Nivel Health Care Consumer Panel: Basic report with information on the panel 2022]. 2022.

12. CBS. Monitor Brede Welvaart & Sustainable Development Goals 2022. Gezondheid [cited 2023 12 June]; Available from: https://www.cbs.nl/nl-nl/dossier/brede-welvaart-en-de-sustainable-development-goals/monitor-brede-welvaart-sustainable-development-goals-2022/verdeling/indicatoren/gezondheid#1

13. Hall, M.A., et al., Trust in physicians and medical institutions: what is it, can it be measured, and does it matter? The milbank quarterly, 2001. 79(4): p. 613-639.

14. Li, T. and H.H. Fung, Age differences in trust: An investigation across 38 countries. Journals of Gerontology Series B: Psychological Sciences and Social Sciences, 2013. 68(3): p. 347-355.

15. Bailey, P.E. and T. Leon, A systematic review and meta-analysis of age-related differences in trust. Psychology and aging, 2019. 34(5): p. 674.

Comment #2: Switching behaviors --- what is the look back period for the switch insurance? I don't know if it is in the past contract period or ever.

Reaction to comment #2: In the Netherlands enrollees can switch health insurance or change their policies from November 12. The last day to decide to discontinue a health insurance policy is December 31. The last day to enroll in a new health insurance is January 31. The new health insurance then starts January 1, with retroactive effect if enrolled to a new health insurance policy in January. As the survey was send out in February 2022, we examined their switching behaviour between November 12, 2021, and January 31, 2022, for the contract period of 2022.

In lines 74-79 we added the following text to the manuscript:

“Every year, from November 12, enrollees have the option of switching to another health insurer or health insurance policy for the coming year. The last day to enroll in a new health insurance is January 31. The new health insurance then starts January 1, retroactively if enrollees enroll in a new health insurance policy in January.” 

Comment #3: Table 7 -- I am unsure how to interpret AGE -- do older people have higher or lower odds of switching? I can't tell direction here. Please clarify.

Reaction to comment #3:

We stated that younger enrollees have switched health insurers slightly more often (OR=0.94, p=0.00). We understand that this may be somewhat confusing, since the OR is related to age in a direction that is increasing. The OR of 0.94 means that older people have a lower odds of switching compared to younger people. Therefore, we have amended this statement on younger enrollees to a statement on older enrollees. The text in lines 338-340 now reads: 

“If we look at the background characteristics, older enrollees have switched health insurers slightly less often than younger enrollees (OR=0.94, p<0.001).”

Comment #4: P=0.00 makes no sense to me, is it better to say p<.01 or p<.0001 as there is some miniscule chance of a random draw from a distribution centered on the null to produce a result as unusual as the ones that are observed.

Reaction to comment #4: 

Thank you for addressing this point. Where applicable, we have changed p=0.00 to p<0.01 or p<0.001 in the manuscript. 

Comment #5: Table 8 hints that there is a rational response to more restrictive insurance by individuals who are less likely to use insurance or who are severely cash-constrained. Is it possible to incorporate income into these analyses to further examine the role of resource availability and choice?

Reaction to comment #5: 

We think this is an interesting point, but unfortunately we do not have information on respondents' income. However, it is a good suggestion to include in follow-up research.

---

## [Decision Letter · Decision Letter 1]

4 Sep 2023

PONE-D-23-04367R1What role does enrollees’ trust in health insurers play when choosing health insurance?PLOS ONE

Dear Dr. Hulst,

Thank you for submitting your manuscript to PLOS ONE. After careful consideration, we feel that it has merit but does not fully meet PLOS ONE’s publication criteria as it currently stands. Therefore, we invite you to submit a revised version of the manuscript that addresses the points raised during the review process.

We look forward to receiving your revised manuscript.

Kind regards,

Tai Ming Wut

Academic Editor

PLOS ONE

Journal Requirements:

Reviewers' comments:

Reviewer's Responses to Questions

**Comments to the Author**

1. If the authors have adequately addressed your comments raised in a previous round of review and you feel that this manuscript is now acceptable for publication, you may indicate that here to bypass the “Comments to the Author” section, enter your conflict of interest statement in the “Confidential to Editor” section, and submit your "Accept" recommendation.

Reviewer #1: (No Response)

2. Is the manuscript technically sound, and do the data support the conclusions?

Reviewer #1: Yes

3. Has the statistical analysis been performed appropriately and rigorously? 

Reviewer #1: Yes

4. Have the authors made all data underlying the findings in their manuscript fully available?

Reviewer #1: Yes

5. Is the manuscript presented in an intelligible fashion and written in standard English?

Reviewer #1: Yes

6. Review Comments to the Author

Reviewer #1: I greatly appreciate the effort that the authors have expended to clarify their interesting and useful work. I have a few remaining comments.

1) I would like a title change to indicate an association. The analysis is, in my opinion, non-causal. It is a very useful descriptive piece that highlights associations of trust and health insurance selection but I am not seeing an RCT or the standard set of quasi-experimental designs (DiD, ITS, regression discontinuities etc). This is recognized in line 452

2) Line 79 “There is a lot to choose from....” Is either not needed or needs to be clarified what THERE refers to.

3) Table 8, please include a precis of each model in the line that includes the model number and N

Other than these three items, I am satisfied with the work and believe it is a useful contribution to the broader scholarly discussion.

7. PLOS authors have the option to publish the peer review history of their article (what does this mean?). If published, this will include your full peer review and any attached files.

Reviewer #1: No

---

## [Author Response · Author response to Decision Letter 1]

18 Sep 2023

Comments from the reviewers: 

Reviewer #1

Comment #1: I would like a title change to indicate an association. The analysis is, in my opinion, non-causal. It is a very useful descriptive piece that highlights associations of trust and health insurance selection but I am not seeing an RCT or the standard set of quasi-experimental designs (DiD, ITS, regression discontinuities etc). This is recognized in line 452.

Reaction to comment #1: The reviewer is right that the analysis may indicate association rather than causation. We therefore agree with the reviewer's request to change the title. We have amended the title of this paper into:

“How is enrollees' trust in health insurers associated with choosing health insurance?”

Furthermore, we noticed that at every point in the manuscript where we described the aim of the study, it was written, as in the title, as examining causation rather than association. We therefore made some additional changes. For example, line 6 described: “This study aims to investigate what role enrollees’ trust in health insurers plays in their choice of a health insurance policy in the Netherlands”. We changed this into:

“This study aims to investigate how enrollees’ trust in health insurers is associated with their choice of a health insurance policy in the Netherlands.” 

Comment #2: Line 79 “There is a lot to choose from....” Is either not needed or needs to be clarified what THERE refers to.

Reaction to comment #2: We agree with the reviewer that this short sentence is unclear and does not add value. We have therefore decided to omit this sentence.

Comment #3: Table 8, please include a precis of each model in the line that includes the model number and N.

Reaction to comment #3: The reviewer is right that it would be clearer to include a precis of each model in Table 8. We have therefore slightly modified the main title of Table 8 and added a precis to each model in the table. For model 2, this is "Examining the association with general trust in health insurers", for model 3, "Examining the association with trust in one's own health insurer" and for model 4, "Examining the association with trust in one's own health insurer's purchasing strategy.”

---

## [Decision Letter · Decision Letter 2]

3 Oct 2023

How is enrollees' trust in health insurers associated with choosing health insurance?

PONE-D-23-04367R2

Dear Frank van der Hulst,

We’re pleased to inform you that your manuscript has been judged scientifically suitable for publication and will be formally accepted for publication once it meets all outstanding technical requirements.

Kind regards,

Tai Ming Wut

Academic Editor

PLOS ONE

Additional Editor Comments (optional):

Reviewers' comments:

Reviewer's Responses to Questions

**Comments to the Author**

1. If the authors have adequately addressed your comments raised in a previous round of review and you feel that this manuscript is now acceptable for publication, you may indicate that here to bypass the “Comments to the Author” section, enter your conflict of interest statement in the “Confidential to Editor” section, and submit your "Accept" recommendation.

Reviewer #1: All comments have been addressed

2. Is the manuscript technically sound, and do the data support the conclusions?

Reviewer #1: Yes

3. Has the statistical analysis been performed appropriately and rigorously? 

Reviewer #1: Yes

4. Have the authors made all data underlying the findings in their manuscript fully available?

Reviewer #1: Yes

5. Is the manuscript presented in an intelligible fashion and written in standard English?

Reviewer #1: Yes

6. Review Comments to the Author

Reviewer #1: (No Response)

7. PLOS authors have the option to publish the peer review history of their article (what does this mean?). If published, this will include your full peer review and any attached files.

Reviewer #1: No

---

## [Editor Report · Acceptance letter]

24 Oct 2023

PONE-D-23-04367R2 

How is enrollees' trust in health insurers associated with choosing health insurance? 

Dear Dr. van der Hulst:

I'm pleased to inform you that your manuscript has been deemed suitable for publication in PLOS ONE. Congratulations! Your manuscript is now with our production department. 

Kind regards, 

on behalf of

Dr. Tai Ming Wut 

Academic Editor

PLOS ONE